

# PermQRDroid: Android malware detection with novel attention layered mini-ResNet architecture over effective permission information image

Kazım Kılıç[1], İbrahim Alper Doğru[1] and Sinan Toklu[1,2]

[1] IoTLab, Department of Computer Engineering, Gazi University, Ankara, Turkey
[2] Information Technology Faculty, Mingachevir State University, Mingeçevir, Mingeçevir, Azerbaijan

Corresponding author
Kazım Kılıç, kazim.kilic@yobu.edu.tr

## ABSTRACT

**Background**. The Android operating system holds the vast majority of the market share in smart device usage worldwide. The Android operating system, which is of interest to users, is increasing its usage rate day by day due to its open source nature and free applications. Applications can be installed on the Android operating system from official application markets and unofficial third-party environments, which poses a great risk to users' privacy and security.

**Methods**. In this study, an attention-layered mini-ResNet model is proposed, which can detect QR code-like images created using the 100 most effective defined permission information of Android applications. In the proposed method, permission information is obtained from four different datasets with different number of applications. QR code-like images of size 10x10x1 are created by selecting effective permissions using the chi-square technique. In the proposed classification architecture, residual layers are used to avoid ignoring the residual features of the images, and attention layers are used to focus on specific regions after each residual layer. The proposed architecture has a low number of parameters and memory consumption despite adding the residual layer and the weighting operations in the attention layer.

**Results**. Using the proposed method, accuracy values of 96.95%, 98.34%, 98.33% and 100% were achieved, respectively, on four datasets containing applications obtained from different sources such as Androzoo, Drebin, Genome and Google Play Store. On the Mix dataset, which is a combination of four datasets, an accuracy value of 96.7% was produced with the proposed method. When 10-fold cross validation was applied to reduce the suggested bias, accuracy values of 97.50%, 98.62%, 98%, 94% and 97.61% were obtained, respectively. The success and durability of the proposed method in different environments have been tested through experiments conducted on different datasets. The results show that the proposed method exhibits better classification performance compared to classical machine learning algorithms, deep learning-based studies using permission information, and similar image-based studies.

# INTRODUCTION

Nowadays, smart devices have become an important part of our lives. The use of smart devices that can perform many tasks that computers can do is increasing, and they make our lives easier thanks to their constantly developing features. It is estimated that there will be 4.88 billion smartphone users worldwide by the end of 2024 (*Turner, 2024*). Approximately 70% of these users prefer Android operating system devices (*Turner, 2024*). There are approximately 2.5 million applications offered to users in the Google Play Store in January 2024 (*Statista, 2024*). The popularity of the Android operating system and its open source code causes security problems. The fact that it has a 70% user rate all over the world makes the Android operating system a target of cybercriminals who write malicious code.

Over time, cyber criminals develop applications coded with different techniques that can run on the Android operating system and inject these applications into the Google Play Store and third-party environments (*Smmarwar, Gupta & Kumar, 2024*). This situation causes malicious applications to become a security problem in the Android operating system. A security problem is when a running system becomes vulnerable to unwanted interventions, data loss, unauthorized access to the system, and other security threats. The main security problems in the Android operating system can be shown as malware, hacking, personal information theft, data leakage, phishing, pretexting, man-in-the-middle attack, denial of service attack, and zero-day attack. Especially malicious software puts users in a difficult situation regarding the theft of sensitive information and data loss. Google developed Google Bouncer and Google Play Protection to block malware (*Mahindru et al., 2024*). But these applications are insufficient to detect malware. 6,463,413 mobile malicious applications were detected by Kaspersky in 2022 (*Shıshkova, 2022*). However, Android applications downloaded from third-party environments do not undergo any security tests and can be installed on the operating system when the user gives permission. This situation leaves users vulnerable.

The Android operating system allows users to allow applications during the installation and use of applications on their devices. Malicious apps tend to request a lot of permissions. However, Android users often ignore the permissions requested by applications, and this poses a risk for users. The permissions requested by applications play an important role in detecting malware (*Seyfari & Meimandi, 2024*). Permission requests of Android applications are defined in the AndroidManifest.xml file. Researchers are interested in machine learning-based android malware detection using permission information.

There are many studies in the literature that use permission information and produce successful results (*Rafiq et al., 2022*; *Seyfari & Meimandi, 2024*). Feature vectors containing "0" and "1" are formed with the permission information obtained in the AndroidManifest file. By using all of this information or selecting effective permissions, Android malware can be detected at a high rate. Deep learning-based approaches, which have been popular in recent years, are also used on permission information for the detection of Android malware (*Fu et al., 2024*). Convolutional neural networks (CNNs) the popular architecture of deep learning, produce successful results in image analysis (*Yadav et al., 2022*). For this

reason, researchers create images using features of Android applications such as byte array or opcode (*Tang et al., 2024*). Researchers who apply transfer learning architectures on these images achieve successful results with the feature extraction and classification power of CNNs. However, the images used in these methods contain a lot of meaningful and meaningless information, and the CNN architectures used also contain a large number of parameters and require high memory consumption.

In this study, a mini-ResNet model with attention layer is presented through QR code-like images based on selected permission information to detect Android malware. The aim of the study is to detect Android malware with a fast, low-parameter and high-accuracy efficient detection model using images of meaningful features. In the proposed model, defined permission information of the applications is obtained and 100 effective permission information on the created dataset are selected using the chi-square technique. QR code-like images of $10 \times 10$ size are created with the selected features. A low-parameter CNN architecture consisting of 5 blocks with residual layers and attention layers is used for feature extraction and classification from these images. The performance of the selection of effective permissions is compared with $20 \times 20$ permission information images consisting of all permissions. For the data independence and validity of the proposed method, experiments are carried out on 4 different datasets and with the Mix dataset, which is a combination of all of them.

The remaining sections of this research paper are organized as follows: The 'Literature Review' section presents Android malware analysis types, summary of permission-based and image-based similar studies. The 'Materials and Methods' section explains the datasets used, the extraction of permission information, the image creation process, and details of the classification architecture. In the 'Results section, measurement values and graphical representations of the test sets of the datasets of the proposed method are presented. In the 'Discussion' section, comparisons are made with similar studies. The 'Conclusion' section gives information about the results observed in this study and future studies.

## Motivation and contributions

The open source nature and popularity of the Android operating system make it a target for cyber attackers. The fact that the Android operating system allows applications downloaded from third-party environments as well as official application markets to be run leaves users vulnerable to cyber attackers. In recent years, many studies have been conducted on Android malware detection. Studies conducted using static analysis are especially popular because they analyze the application before it is run and because they are fast. Information obtained from the application file is needed for malware detection using static analysis. In Android applications, the permission information requested from the user is of critical importance. These permissions are effective in deciding whether the applications are malware or benign. Especially malware applications tend to demand more work. For this reason, researchers have conducted many studies focusing on permission information. Permission information vectors consist of values of 0 and 1. Not all permission information may be meaningful for classification. For this reason, permission information is selected

using feature selection techniques in the studies conducted and classified using classical machine learning methods.

In recent years, hardware developments have increased the interest in deep learning architectures. Android malware researchers are interested in classifying the features obtained from Android applications using architectures such as one-dimensional convolutional neural network model (1DCNN) and long short term memory (LSTM) in order to obtain more successful results. At the same time, they provide image transformation by digitizing the byte array or textual information of the applications. Successful results have been obtained by using two-dimensional convolutional neural networks (2DCNN) and transfer learning approaches. However, there is a lot of meaningless information in the studies where image transformation is performed. The created image sizes are large and the CNN architectures used contain a large number of parameters. Lightweight and low-parameter structures are insufficient in terms of performance. Some studies have used residual blocks and attention blocks to increase performance, but the methods they present contain a large number of parameters. In addition, studies on Android malware detection have been performed on a single dataset. The performance of the proposed models on different datasets has not been shown.

In this study, permission information that is effective for Android malware detection is selected and converted into QR code-like images. The aim is to both use the effect of permission information and reduce the number of parameters with small-sized images created from permission information. It is aimed to benefit from the feature extraction and classification power of CNNs by not using meaningless information in the image and ensuring image transformation of effective permission information. The proposed method is tested on four data sets obtained from different sources. At the same time, both the bias of the proposed model is reduced and the effect of the established architecture is measured with cross validation and ablation studies.

The following research questions have been determined in line with our aims and objectives:

RQ1 What is the classification performance when the permission information of Android applications is converted to images? In this context, 20x20 images were created and classified without subjecting the permission information of the applications to any feature selection process.

RQ2 Can detection performance be increased by using small-sized images created by selecting effective permissions and eliminating meaningless information? The most effective 100 features were selected using the chi-square technique and 10x10 images were created.

RQ3 What is the performance of the lightweight CNN architecture to be created using the attention block and residual block against existing methods? The findings obtained with the created architecture are compared with existing methods.

RQ4 What is the performance of the proposed model on data sets with multiple and different data numbers, unlike existing studies? The performance of the proposed model was tested on four different data sets. At the same time, all data sets were combined to

create a Mix dataset. The consistency of the proposed model was measured using cross validation for all data sets.

RQ5 Is the performance of the proposed model with images generated from permission information successful against classical machine learning techniques? Classification was performed with classical machine learning techniques and compared with the proposed model.

The main contributions of our study to the literature, which was conducted as a result of experiments to seek answers to the five research questions above, can be summarized as follows:

- In this study, four data sets from different sources were used to verify the proposed method. Similar studies have conducted experiments on only one data set. The stability and reliability of the proposed model, independent of the number of data, were tested with data sets with different data numbers used in the study.
- There are image-based studies that use permission information, but as seen in the literature review, this study is the first to use a QR code-like image with the selection of effective permissions.
- A new attention layered mini-ResNet architecture with five blocks, including three residual layers and three attention layers, is proposed for feature extraction and classification. The proposed architecture does not ignore residual features with residual layers and achieves high accuracy by focusing on certain parts of the image with attention layers in each block.
- The proposed architecture has the advantage of being able to work with images of minimum $10 \times 10$ size and higher compared to high-resolution studies that use byte array images in Android malware detection.
- The attention layered mini-ResNet model with contains a very low number of parameters and takes up less space in memory compared to transfer learning architectures. It has approximately 11 times fewer parameters and 3.5 times less memory size than MobileNet, the lightest and fastest transfer learning architecture..
- The proposed method produces more successful results compared to similar permission-based and image-based studies in the literature.

## LITERATURE REVIEW

Since the Android operating system is widely used and open source, Android users are at risk of malware. Three different approaches are used to determine whether Android applications are malware or benign: static, dynamic and hybrid.

Static analysis involves examining the malicious code or file without executing it. This approach allows security researchers and analysts to understand the structure, behavior, and potential threats posed by malware without the risk of infecting a system. Researchers using this technique usually analyze zip files with .apk extensions and use classes.dex files together with the AndroidManifest.xml file. The information contained in the xml file (permissions, intentions, activities, *etc.*) is widely used to detect Android malware (*Atacak, 2023*). In the Classes.dex file, malware detection studies using features such as opcode

and API calls have achieved successful results (*Mahindru et al., 2024*). Deep learning and natural language processing techniques are used on features obtained from xml and dex files. *Kabakus (2022)* performed feature extraction with natural language processing methods and classification with 1DCNN on the permission information and API calls of Android applications. The researcher has automated the feature extraction, feature selection and classification processes with the method called DroidMalwareDetector (*Kabakus, 2022*). *Khan et al. (2022)* presented the Op2vec method, which converts the opcodes of applications into digital vectors using natural language processing techniques and classifies them based on deep learning. In recent years, the development of large language models has made transformer architecture popular. Detecting Android malware from xml and dex file information using Transformer is among the current issues (*Saracino & Simoni, 2023*; *Rahali & Akhloufi, 2021*).

In the dynamic analysis method, the application is executed in a controlled environment to observe the behavior of the malicious code and understand its capabilities. The dynamic analysis approach is time-consuming and laborious. However, compared to static analysis, it has an advantage against code obfuscation techniques and polymorphic software (*Smmarwar, Gupta & Kumar, 2024*). In the dynamic analysis approach, researchers generally focus on system calls and network traffic data (*Alomari et al., 2023*). *Xiao et al. (2019)* proposed an LSTM-based classifier that detects system call sequences of applications by detecting them as sentences.

Hybrid analysis involves the combined use of information obtained from static analysis and dynamic analysis. The features obtained from the xml file or dex file through static analysis and the features extracted as a result of dynamic analysis are combined. It is powerful as it includes the advantages of static and dynamic approaches in terms of detection, but it also has the disadvantage of requiring time and expert knowledge (*Aurangzeb & Aleem, 2023*).

### Studies using permission information

*Atacak, Kılıç & Doğru (2022)* proposed a new feature reducer and classifier CNN+Adaptive Neuro-Fuzzy Inference System (ANFIS) hybrid model for the detection of Android malware. In their proposed method, they used convolution and fully connected layer instead of feature selection methods such as information gain from permission information and chi-square test. By obtaining new features from the permission information with convolution layers, they integrated the data as input into the ANFIS architecture with five neurons placed in the fully connected layer. They tested their CNN+ANFIS architecture with two different datasets. They achieved 92% accuracy in the first dataset and 94.66% accuracy in the second dataset.

*Mat et al. (2022)* tried to detect Android malware using the Bayesian probability algorithm. They extracted permission information from 10,000 apk files obtained from Androzoo and Drebin datasets. They applied information gain and chi-square test to select permission information and presented comparative analysis with different number of features. The most successful result was found to be 91.1% by selecting 30 features with the chi-square test.

*Altaher & Barukab (2017)* proposed a fuzzy logic based classifier on permission information. They obtained the permission information of the applications and selected 24 features using the information gain method. In the classification stage, they used the hybrid fuzzy c-means (FCM)-ANFIS model by combining fuzzy c-means clustering and ANFIS model. As a result of the study, they achieved 91% accuracy (*Altaher & Barukap, 2017*). In a similar study, *Abdulla & Altaher (2015)* extracted the permission information of 200 applications. They obtained 24 features with the information gain method. They divided the $1 \times 24$ feature vector consisting of 0 and 1 values into 3 groups and converted the 8 values in each group into byte format. In the classification stage, they used KNN-based fuzzy clustering method together with ANFIS. As a result of the study, they reached an accuracy value of 75% (*Abdulla & Altaher, 2015*).

*Arslan (2022)* tried to detect malware by dividing permission-based features into groups of 11 permissions. The unbalanced dataset was balanced using the SMOTE technique. He conducted experiments with classical machine learning techniques, DNN, LSTM and GRU for the classification process. As a result of the experiments, it was observed that the most successful result was obtained with the ExtraTree algorithm and reached 92.9% accuracy with this method (*Arslan, 2022*).

*Şahin et al. (2023)* proposed a linear regression-based method on permission information. They tested their methods on different datasets and compared them with different classifiers. Researchers who increased the classification performance with ensemble classifiers achieved an accuracy value of 95.6% with the AMD (*Wei et al., 2017*) dataset and 91.87% with the Lopez's (*Urcuqui-López & Cadavid, 2016*) dataset. They reached an accuracy value of 82.94% with M0Droid (*Damshenas et al., 2015*) and 96.69% with Arslan's (*Arslan, 2021*) dataset (*Şahın, Akleylek & Kiliç, 2022*). In a similar study, *Şahin et al. (2023)* aimed to eliminate unnecessary features by using a linear regression-based feature selection approach during the feature selection phase. In the experiments, they obtained a 96.1% F-score with the MLP algorithm (*Şahin et al., 2023*).

## Image based studies

Successful results are achieved for Android malware detection by using deep learning and machine learning methods with developing GPUs and CPUs. While researchers search for different features of applications, they also convert the features into different formats such as images and audio (*Yadav et al., 2022*; *Tarwireyi, Terzoli & Adigun, 2023*; *Kural, Kiliç & Aksaç, 2023*; *Tang et al., 2024*). The reason for this is to benefit from the feature extraction and classification power of deep learning. However, in studies where image transformation is performed, feature selection is automatically extracted and classification can be made without the need for expert knowledge.

*Yadav et al. (2022)* used dex files of applications for Android malware detection. They converted the byte array of Dex files into a square format image with RGB color channels. Their study presented a comparative analysis using a stacking ensemble classifier consisting of SVM and RF algorithms and the analysis of 26 different pre-trained CNN architectures. Researchers who want to use the advantages of transfer learning in malware detection have achieved an accuracy value of 95.7% on 5,986 images with the EfficientNet -B4 architecture.

*Yen & Sun (2019)* analyzed apk files using natural language processing methods. Researchers weighted the texts with the Tf-IDF method and converted the numerical feature vectors they obtained into images. After analyzing a total of 1,440 applications and creating the image dataset, they classified them using CNN architecture. As a result of their studies, they reached 92% accuracy.

*Xiao & Yang (2020)* developed a CNN architecture that learns from dalvik byte code. In their method, the dalvik byte code stored in dex files is converted into images with RGB channels. They carried out experiments on a total of 10,540 images with the lightweight CNN architecture they established. As a result of the experiment, they obtained an accuracy value of 93%.

*Zhu et al. (2023)* created feature vectors using permissions, hardware and API calls. They converted the feature vector they obtained into an image of $18 \times 18$ size. They presented a new CNN architecture called MSerNetDroid to classify images. They tested their proposed MSerNetDroid architecture on 3,187 applications collected from Virusshare and Google Play Store and reached 96.48% accuracy.

*Aurangzeb et al. (2024)* created the AndroDex dataset containing 24,746 apk files. This dataset, which includes Dex images, also represents apk files with code obfuscation techniques applied. In their study, the researchers applied normalization to the images and then performed dimension reduction with PCA. In their experiments with classical machine learning algorithms, they achieved a 95% accuracy value with the XGBoost algorithm.

*Tasyurek & Arslan (2023)* proposed a method called RT-Droid to detect real-time Android malware. In their proposed method, permission information is extracted from the manifest.xml file and converted to a $19 \times 19$ image in RGB format. 6,760 malignant samples were obtained from Drebin and Genome projects, and 961 benign samples were obtained from Arslan's dataset. They used YOLO V5 architecture to classify images in their fast detection method. Using the transfer learning technique, they reached an accuracy value of 94.2% with the YOLO V5 architecture.

*Arslan & Tasyurek (2022)* proposed the AMD-CNN method for malware detection using graphical representations. In their proposed method, they obtained all the information from the manifest.xml file and converted it into a 2D-code image. In their study consisting of a total of 1920 images, they used CNN architecture for classification and feature extraction. They achieved 96.2% accuracy performance with the AMD-CNN method, which can make fast detection. Table S1 provides a summary of image-based Android malware detection studies.

## MATERIAL AND METHOD

In this section, the datasets used in the study, feature selection and conversion of permission information into a QR code-like image, details of the proposed architecture and the hyper-parameters used are explained.

## Data collection

In order to evaluate our method for Android malware detection, four different datasets were used in this study. These datasets are named Dataset1, Dataset2, Dataset3 and Dataset4. Datasets were collected from different environments and their apk quantities are different. The purpose of this is to demonstrate the effectiveness and data independence of the presented method. In addition, it is to create a robust model against the disadvantages of all of these datasets.

Dataset1 was collected from the Androzoo (*Allix et al., 2016*) environment and contains 20,000 benign and 20,000 malware samples. There are currently 24,499,587 applications in the Androzoo environment. Malware samples belonging to Dataset1 used in this study were detected by at least 10 antivirus programs and are available in the Google Play Store. Benign samples, on the other hand, could not be detected by antivirus programs and were downloaded based on the condition that they were found in the Google Play Store.

Dataset2 contains 5,498 applications from the Drebin (*Arp et al., 2014*) dataset and 1,163 applications from the Genome (*Zhou & Jiang, 2012*) dataset. Benign applications were collected by *Arslan (2021)* and there are 961 pieces. The dataset containing a total of 7,622 applications was used in *Arslan*'s (*2021*) study called "AndroAnalyzer".

Dataset3 consists of 1,000 benign and 1,000 malware samples. Malware samples were taken from the Drebin (*Arp et al., 2014*) dataset. The Drebin dataset, which consists of 179 malware families, contains a total of 5,560 malware. Benign samples were obtained from the APKPure website. Benign samples are available on the Google Play Store market and were scanned with VirusTotal (*VT Team, 2020*).

Dataset4 contains 250 benign and 250 malware samples. Malware samples were taken from the CICMalDroid 2020 (*Mahdavifar et al., 2020*) dataset. The CICMalDroid dataset contains 17,341 applications in five different categories. Benign applications were obtained from the Google Play Store market and scanned with the VirusTotal application and they were determined to be benign. Table S2 gives the number of applications and sources of the datasets.

## Proposed method

In this study, a QR code-like image of permission information is created to detect Android malware. These images are classified with a new CNN architecture that has attention and residual layers. In the first stage of our method, permission information is extracted from the AndroidManifesxt.xml file. After this stage, the 100 most effective permission information is selected with the chi-square test on the created dataset. Single-channel QR code-like images are created after the selected permissions are converted into a $10 \times 10$ matrix. These images contain low-dimensional and meaningful information. Our goal is to detect malware quickly and with high accuracy. In the final stage, a mini-ResNet model with 3 block layers containing residual layers is created for classification. By adding attention layers to this model, PermQRDroid architecture is created. Figure 1 shows the graph of the proposed method.

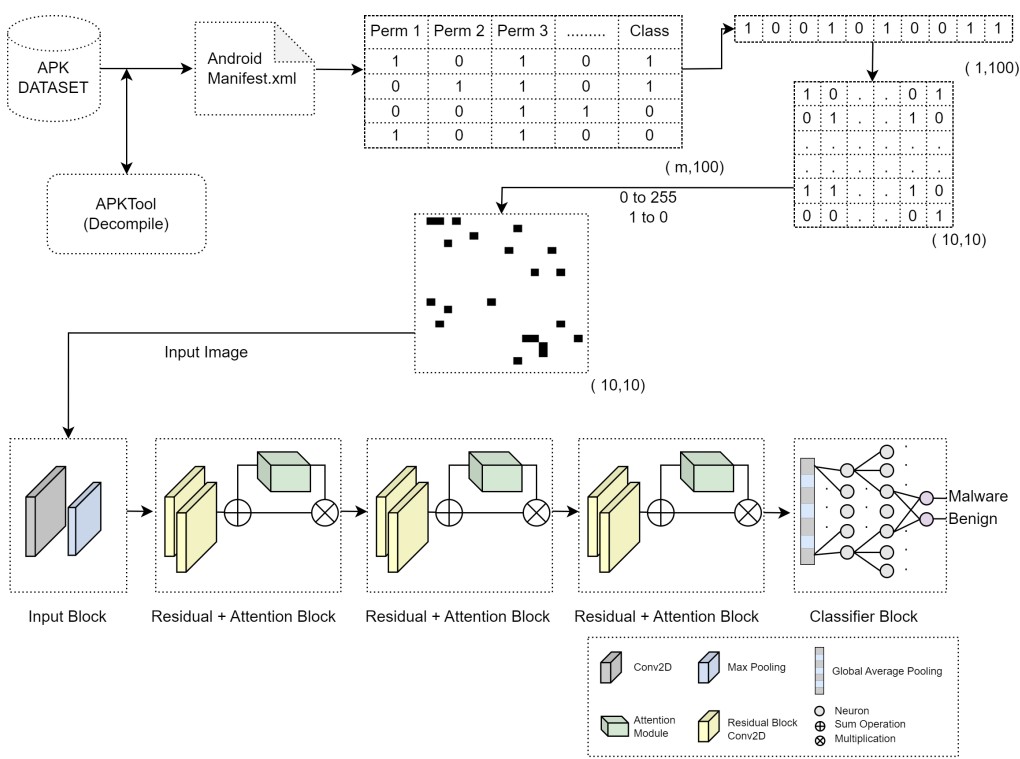

**Figure 1  Flow diagram of the proposed method.** Feature extraction process from applications, feature selection process, image transformation, attention layered mini-ResNet architecture used for classification.

## Feature extraction and selection process

In detection studies using static analysis, the first step is usually to decompile files with .apk extension. APK (Android Pocket Kit) are compressed .zip type files that contain libraries, source codes, permissions, directories and resources of Android applications (*Aurangzeb et al., 2024*). In the feature extraction process, firstly, apk extension files were analyzed using APKTool and AndroidManifest.xml files were obtained. Permission information was obtained from xml files kept with application names in a single folder. Two approaches can be followed to obtain permission information and create feature vectors. First, a list of known permission names can be created in the Android operating system and checked for each application whether it contains a permission in the permission list. In the second approach used in this study, the manifest files of existing apk files are accessed and the permissions requested by each application are kept in a list. For example, if more than one application requires the same permission, that permission information is written once. However, when a different permission is seen, it is added to the list. The columns of the Excel file are created with the created list, and then for each application, the value 1 is added to the permission field it contains, and 0 is added to the permission fields it does

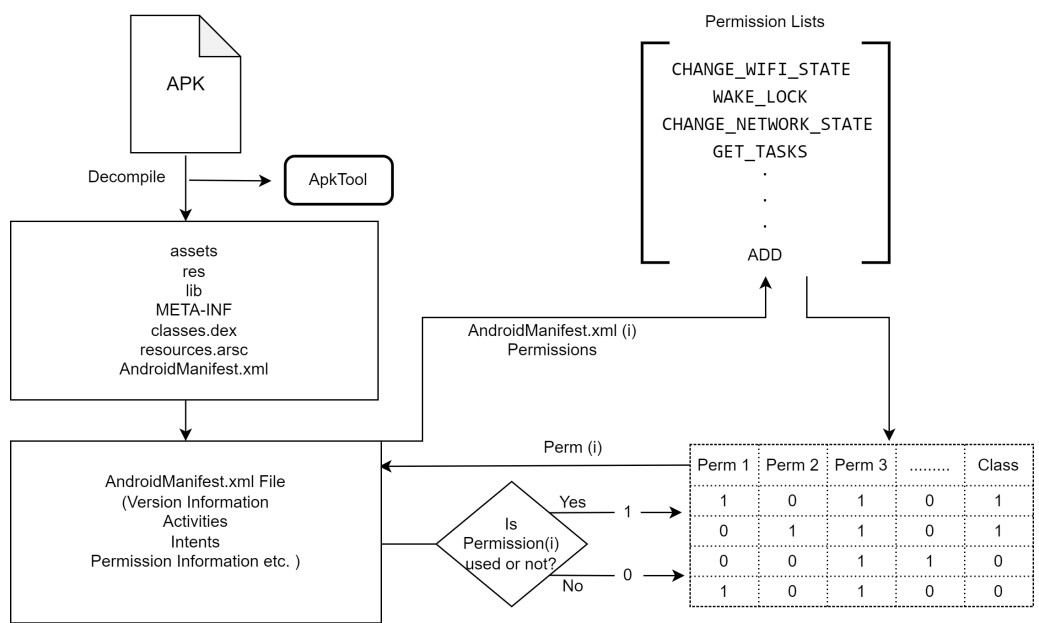

**Figure 2** Permission information extraction process and feature vector creation.

not contain. The process of obtaining permission information from the manifest file and creating feature vectors is shown in Fig. 2.

Feature selection is the process of eliminating irrelevant features and selecting effective features that have a high impact on classification. This process has proven effective in detecting Android malware, especially in studies using permission information (*Altaher & Barukap, 2017*; *Abdulla & Altaher, 2015*; *Şahin et al., 2023*). It has been determined that some information in the list of permission information has no effect on classification. For this reason, chi square technique was used to obtain 100 effective features. Selection of the effective 100 features both reduces the parameter of the model and increases the detection speed.

Chi-square Method: This test, which is a statistical method, is used in the analysis of categorical data. It evaluates the difference between real frequencies and desired frequencies and checks the accuracy of a hypothesis accordingly (*Dhal & Azad, 2022*). Chi-square test is calculated using the following formula.

$$\chi^2 = \sum_{i=1}^{k} \frac{(O_i - E_i)^2}{E_i}.$$   (1)

In this formula, $O_i$ refers to the observed values, $E_i$ refers to the expected values, and $k$ refers to the number of categories.

Chi-square test is applied to the dataset consisting of permission information and the classification effect scores of all features are calculated. According to the score, all features are ranked from largest to smallest. Starting from the highest, 100 permission information

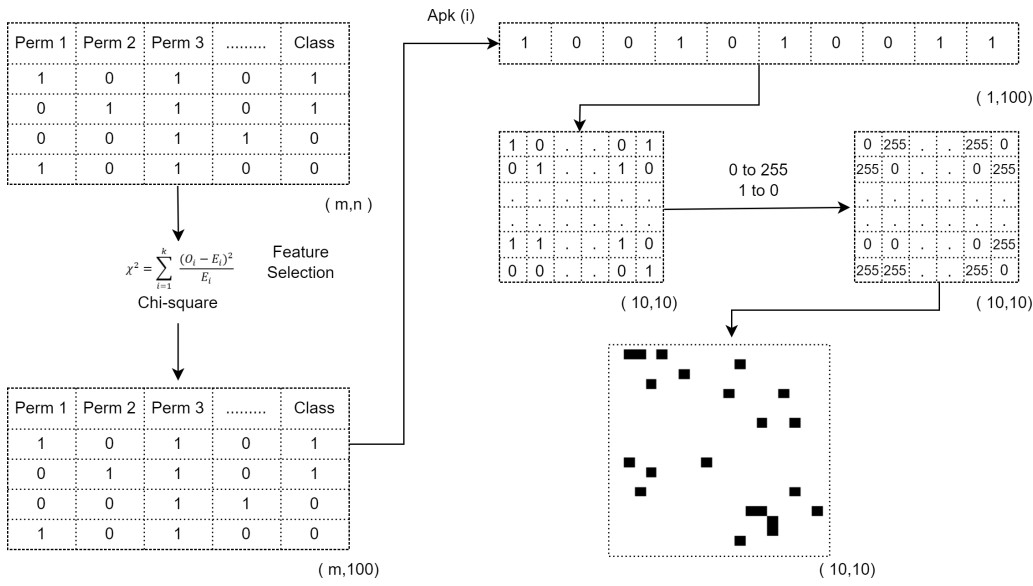

**Figure 3** **Image creation process.**

is selected and other permission information is removed from the dataset. SkiLearn library in Python programming language was used for chi-square test.

## Image creation process

The permission information extraction approach applied in this study may differ depending on the dataset and number of applications. Four different datasets are used for experiments. The permission information vectors extracted in these datasets consist of feature vectors in the range of $1 \times 325$–$1 \times 398$. These values are not suitable for creating a square image. CNN architectures, on the other hand, can work with square format images. For this reason, raw permit information is first converted into a feature vector of size $1 \times 400$ for each dataset and square matrices of size $20 \times 20$ are obtained. For conversion to $1 \times 400$ size, the value 0 is added to the feature vector as many as the number of missing columns, ensuring that it becomes a perfect square.

The experiments are evaluated on two different image sizes. One of the main aims of the study is to make detection quickly and with high accuracy through images created by effective features. For this reason, 100 effective permission information is selected and 10 $\times$ 10 square matrices are obtained. A value of 255 is assigned to each 0 in the matrices to transform the $10 \times 10$ and $20 \times 20$ sized matrices of the datasets into images, and a 0 value is assigned to each 1 value in the matrices to create QR code-like images. The images created have a single channel of size $10 \times 10 \times 1$ and $20 \times 20 \times 1$. The diagram of

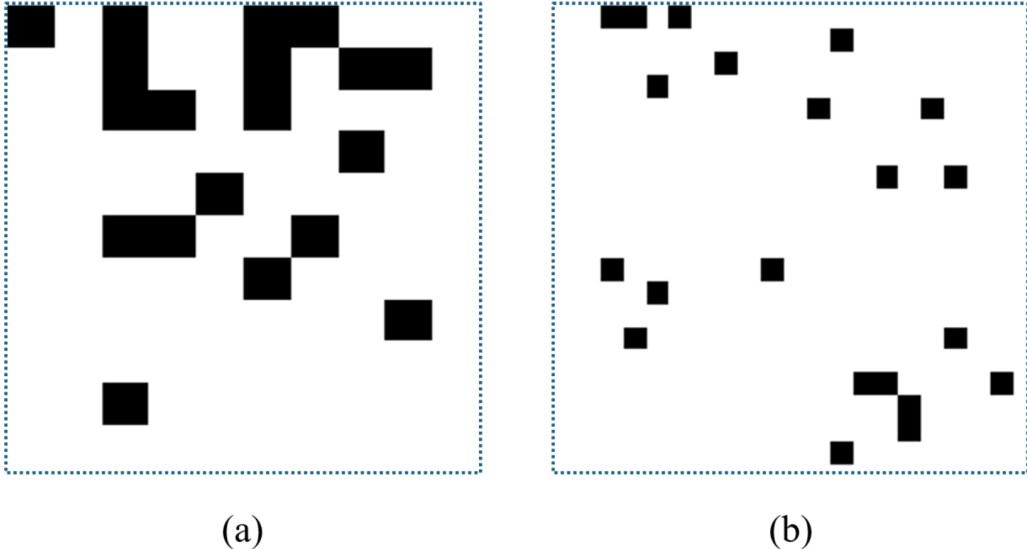

(a)                     (b)

**Figure 4** **QR code-like images (a) 10 × 10 × 1 image, (b) 20 × 20 × 1 image.**

square matrix formation from permission information vectors and QR code-like image transformation is shown in Fig. 3.

QR code-like images created in $10 \times 10 \times 1$ and $20 \times 20 \times 1$ dimensions are shown in Fig. 4.

The selection of 100 permission information and creation of single-channel QR code-like images of $10 \times 10 \times 1$ size will ensure that the number of parameters of the network is low in both the feature extraction and classification stages. In addition, the created CNN architecture will also positively affect the detection speed and ensure that memory consumption is less than a 3-channel image.

## Model architecture

Deep learning, which essentially emerges by deepening artificial neural networks, enables complex calculations to be made on large amounts of data (*LeCun, Bengio & Hinton, 2015*). In the task of feature extraction and classification from images, CNN are popular and are the most widely used deep learning architecture. While CNN architectures can process images using 2D convolution layers, they can also work on 1D data with 1D layers (*Atacak, Kılıç & Doğru, 2022*). Classic CNN architecture consists of convolution, activation, pooling and fully connected layers (*Gu et al., 2018*). In the convolution layer, a filter is moved over the image matrix with the specified size and the specified number of steps. In this shifting process, the weights on the filter are multiplied by the values in the image matrix and a new value is obtained from the sum of all multiplications (*Li et al., 2021*). The convolution process is calculated with the following formula.

$$Z_{i,j}^{(k)} = (X * W^{(k)})_{i,j} + b^{(k)} = \sum_{m=0}^{M-1}\sum_{n=0}^{N-1} X_{i+m,j+n} \cdot W_{m,n}^{(k)} + b^{(k)}.$$
(2)

In this formula, X is the input information, W stands for the weight matrix, b stands for the bias value, * stands for the convolution operator, M and N stand for the filter size, and Z stands for the output. An output is created in the activation layer, which takes each value in the feature map obtained from the convolution layer as input. The output value of the ReLU activation function is calculated with the formula below.

$$A_{i,j}^{(k)} = ReLU(Z_{i,j}^{(k)}) = max(0, Z_{i,j}^{(k)}). \tag{3}$$

In this formula, $A$ refers to the output of the activation layer and $Z$ refers to the output of the convolution layer.

In the pooling layer, the size of the feature maps is reduced. It allows capturing important features of the current image and allows representing the image in a smaller size. Maximum pooling is calculated by the formula below.

$$P_{i,j}^{(k)} = max_{(m,n) \in pool} A_{i+m,j+n}^{(k)}. \tag{4}$$

Here P refers to the output value of the pooling layer. "max" refers to selecting the maximum value in the pooling region, and "pool" refers to the window dimensions.

## Residual block

Residual blocks used in the ResNet architecture introduced by *He et al. (2016)* were presented as a solution to the vanishing gradient problem in CNN networks. Residual blocks prevent residual feature from being ignored by adding the input information to the output information of the layers (*He et al., 2016*). Transactions made in residual blocks are calculated with the following formula.

$$y = F(x, \{W_i\}) + x. \tag{5}$$

In this formula, x refers to the input information and Wi refers to the weight values in the filters. According to the formula, the weights are multiplied by the x input information and feature maps are created. By adding x input information to these feature maps, y output is obtained.

## Attention block

Attention blocks enable focusing on the prominent features of the image whose features are extracted in CNN architectures (*Vaswani et al., 2017*; *Niu, Zhong & Yu, 2021*). This block consists of two stages: squeezing and excitation blocks.

**Squeeze:** In this process, global average pooling is performed and compression is performed on a channel basis. The formula for the compression process is as follows.

$$z_c = \frac{1}{H \times W} \sum_{i=1}^{H} \sum_{j=1}^{W} X_{i,j,c}. \tag{6}$$

In this formula, X refers to the input information and its dimensions are expressed as HxWxC. HxW is the height and width of the input matrix. C refers to the number of channels and $z_c$ refers to the compressed features for channel c.

**Excitation:** Calculates the weight for each channel in the compression phase. For this process, $1 \times 1$ weights are used in a convolution layer and a sigmoid activation function is applied to the obtained values. The formula for the excitation process is as follows.

$$X = \delta(z_c) \tag{7}$$

$$z_c = \frac{1}{H \times W} \sum_{i=1}^{H} \sum_{j=1}^{W} X_{i,j,c} \tag{8}$$

$$s = \sigma(z_c). \tag{9}$$

Here: $\delta$ is the ReLU activation function, $\sigma$ is the sigmoid activation function and $s$ is the weight vector of the channels. After the compression and excitation processes, the weights calculated for each channel are multiplied by the input information. This process is called scaling. The formula for the scaling process is as follows.

$$X'_{i,j,c} = s_c X_{i,j,c}. \tag{10}$$

In this formula, $X'$ refers to the information appearing in the attention block, and $s_c$ refers to the weights calculated for the relevant channel.

### Attention layered mini-ResNet model

An attention layer mini-ResNet model that can work with $10 \times 10$ sized images was created to detect Android malware with permission information-based QR code-like images. In addition to the classical CNN architectures, the proposed architecture includes residual blocks and an attention layer after each residual block. The proposed architecture consists of five blocks. The first block contains the classic convolution layer, ReLu activation layer and MaxPooling layer, which contains 32 filters of size $3 \times 3$. In the second block, third block and fourth blocks, the residual layer and the attention layer are located together, respectively. Residual layers consist of two convolution layers using 32 filters and a $3 \times 3$ kernel. Attention layers come immediately after residual blocks. In attention layers, $1 \times 1$ sized convolution layers and ReLU are used in the compression process. The activation function used in the excitation process is sigmoid. At the end of the fourth block, there is a global pooling layer and 2D data is made available for the fully connected layer. In the fifth block, there are two fully connected layers and a softmax layer to classify the extracted features. Dropout layers with a ratio of 0.25 were used to prevent overfitting of the network between the fully connected layers. Details of the proposed architecture are shown in Fig. 1.

### Training procedure

Four different datasets were used to evaluate the proposed classification model. $10 \times 10$ and $20 \times 20$ images were created for all datasets. Each dataset is divided into 70% training, 15% validation and 15% testing in the experimental phase.

Training of CNN architectures is done in two ways: the first is training from scratch and for this, weight initial values must be determined, and the second is the transfer learning approach. Since the proposed CNN architecture is not suitable for transfer learning,

Xavier Initialization was used to initialize the weights. The loading and random loading options were also evaluated, but since they made the network learn late and unsuccessful results were obtained, it was decided to use Xavier loading. In the proposed mini-ResNet architecture with attention layer, ReLU is used for activation in the convolution layers, Sigmoid function is used for stimulation in the attention layer, and Softmax function is used in the classifier layer. The network's learning rate started at 0.001, and the learning rate was reduced by 0.2 at the validation loss value, which did not decrease for five epochs. The minimum learning rate was determined as 0.00001. Adam was used for the optimization of the network and binary_crossentropy was used for the loss function. In the training phase, the number of epochs is 20 and the batch size is 64. The hyperparameters used in the original ResNET architecture were used to create the attention layered mini-ResNet architecture. The depth of the network (residual and attention layer numbers) was adjusted according to the image size. An attempt was made to obtain a minimum depth that could work on 10x10 images and contained low parameter.

## RESULTS

This section presents the proposed attention layer mini-ResNet model's performance on permission information images. Experiments were carried out on four different datasets. These datasets are named Dataset 1, Dataset 2, Dataset 3 and Dataset 4 under the heading "3.1.Datasets". The combination of all datasets is called Mix Dataset. The permission information obtained from the used datasets was converted into QR-like images of 10 × 10 and 20 × 20 sizes. Images with a size of 10 × 10 consist of 100 effective features selected using the chi-square technique. In order to obtain a 20 × 20 image, a value of 0 was added to the permission information obtained from the datasets as long as the missing column information. For model performance, all datasets are divided into 70% training, 15% validation and 15% testing. Table 1 presents the validation and test accuracies of the proposed model on Dataset 1, Dataset 2, Dataset 3 and Dataset 4. The performance of the model was evaluated using precision, recall, F-score and accuracy metrics. At the same time, the classification successes obtained from 10 × 10 and 20 × 20 sized images are compared in Table 1.

Underlined and bolded values in Table 1 indicate the accuracy values obtained with the proposed model on the test sets of the datasets. All datasets except Dataset 2 are balanced and therefore their accuracy values are evaluated in performance measurement. As a result of the experiments, test accuracy values of 0.9695, 0.9834, 0.9833 and 1.0000 were achieved for datasets on 10 × 10 images, respectively, with the proposed architecture. On 20 × 20 images, accuracy values of 0.9661, 0.9755, 0.9667 and 0.9733 were obtained, respectively. According to the results obtained, the classification results of 10 × 10 sized images are more successful than the 20 × 20 sized images. Using 10 × 10 and 20 × 20 images, an accuracy value of over 96.60% was achieved in all datasets. The most successful classification result is 100% accuracy on 10 × 10 images in Dataset 4, and the least successful result is 96.61%

**Table 1 Performance of the proposed model on 10 × 10 and 20 × 20 images.** Underlined and bolded values indicate the accuracy values obtained with the proposed model on the test sets of the datasets.

| | | 10 × 10 Image | | | | 20 × 20 Image | | | |
|---|---|---|---|---|---|---|---|---|---|
| | | Pre | Rec | F-sc | Acc | Pre | Rec | F-sc | Acc |
| Dataset 1 | Valid | 0.9606 | 0.9603 | 0.9603 | 0.9603 | 0.9652 | 0.9650 | 0.9650 | 0.9650 |
| | Test | 0.9697 | 0.9695 | 0.9695 | **0.9695** | 0.9664 | 0.9661 | 0.9661 | **0.9661** |
| Dataset 2 | Valid | 0.9761 | 0.9764 | 0.9759 | 0.9764 | 0.9769 | 0.9773 | 0.9769 | 0.9773 |
| | Test | 0.9832 | 0.9834 | 0.9832 | **0.9834** | 0.9752 | 0.9755 | 0.9753 | **0.9755** |
| Dataset 3 | Valid | 0.9839 | 0.9833 | 0.9833 | 0.9833 | 0.9800 | 0.9800 | 0.9800 | 0.9800 |
| | Test | 0.9839 | 0.9833 | 0.9833 | **0.9833** | 0.9687 | 0.9667 | 0.9666 | **0.9667** |
| Dataset 4 | Valid | 0.9484 | 0.9467 | 0.9468 | 0.9467 | 0.9104 | 0.9067 | 0.9071 | 0.9067 |
| | Test | 1.0000 | 1.0000 | 1.0000 | **1.0000** | 0.9733 | 0.9733 | 0.9733 | **0.9733** |
| Mix dataset | Valid | 0.9598 | 0.9597 | 0.9597 | 0.9597 | - | - | - | - |
| | Test | 0.9680 | 0.9678 | 0.9678 | **0.9678** | - | - | - | - |

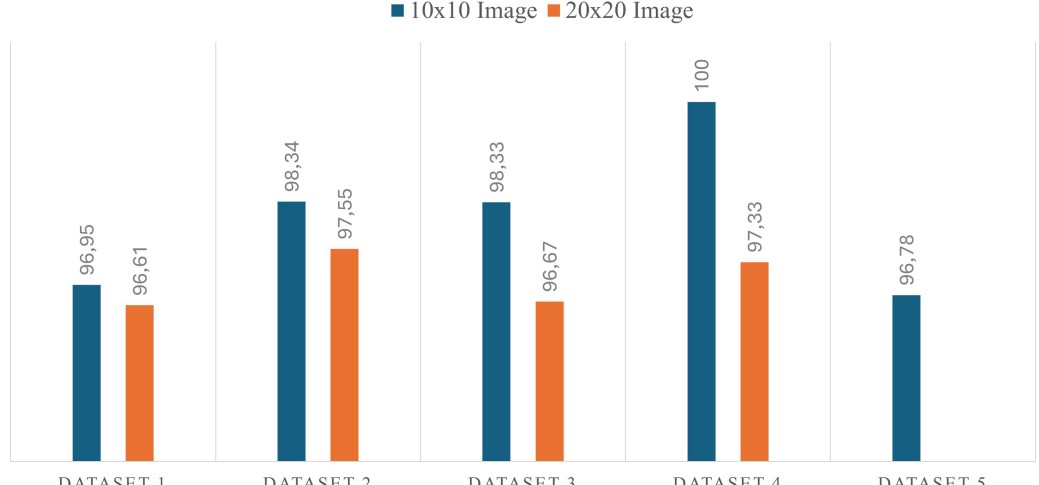

**Figure 5 Accuracy graph of the proposed model on datasets.**

on 20 × 20 images in Dataset 1. Figure 5 shows the accuracy values of the proposed architecture for datasets obtained with 10 × 10 and 20 × 20 images.

This study proposes 10 × 10 QR-like images created from effective permission information selected using the chi-square technique for Android malware detection. The loss curves and confusion matrices shown in Figs. S1 and S2 show the performance of the proposed classification architecture on 10 × 10 sized images. ROC curves showing the ability of the proposed classification model to distinguish two classes on all datasets are shown in Fig. S1.

In order to reduce bias and test the reliability of the classification model, 10-fold cross validation technique was applied on all data sets. Standard deviation (Std) and confidence intervals (Ci) were calculated according to the accuracy values obtained in each fold for all

**Table 2  Accuracy values obtained with cross validation technique.**

|  | Pre | Rec | F-sc | Acc | Std | Ci |
|---|---|---|---|---|---|---|
| Dataset 1 | 0.9765 | 0.9762 | 0.9762 | 0.9750 | 0.0056 | [0.9710, 0.9782] |
| Dataset 2 | 0.9873 | 0.9874 | 0.9872 | 0.9862 | 0.0057 | [0.9824, 0.9895] |
| Dataset 3 | 0.9837 | 0.9834 | 0.9834 | 0.9800 | 0.0080 | [0.9745, 0.9845] |
| Dataset 4 | 0.9426 | 0.9420 | 0.9419 | 0.9400 | 0.0334 | [0.9200, 0.9600] |
| Mix dataset | 0.9776 | 0.9774 | 0.9774 | 0.9761 | 0.0039 | [0.9731, 0.9781] |

**Table 3  Ablation study.**

|  | Basic CNN | Mini-ResNet | Attention Layered Mini-ResNET |
|---|---|---|---|
| Dataset 1 | 0.9550 | 0.9640 | 0.9695 |
| Dataset 2 | 0.8627 | 0.9737 | 0.9834 |
| Dataset 3 | 0.9700 | 0.9733 | 0.9833 |
| Dataset 4 | 0.9600 | 0.9733 | 1.0000 |
| Mix dataset | 0.9497 | 0.9647 | 0.9678 |

data sets. Table 2 shows the average accuracy values, standard deviation and confidence intervals obtained using the cross validation technique on the data sets used in the study.

As a result of cross validation, it is seen that the proposed model is more successful and stable on Dataset 1, Dataset 2 and Mix Dataset, which have a high number of data. However, the result is less successful on Dataset 3 and Dataset 4, which have a low number of data compared to other datasets. The results obtained on Dataset 3 and Dataset 4 are similar to the results in Table 1, where 15% test set is used, and show that the proposed model is highly successful. The findings obtained as a result of cross validation show that the standard deviation value of the proposed model is low when the number of data is high, and the standard deviation value increases when the number of data decreases. However, the standard deviation values obtained with the proposed model are low, and the accuracy values found for each fold are close to each other.

## Ablation study

In this study, instead of the basic CNN architecture, a lightweight CNN architecture with residual and attention layers is proposed. In order to demonstrate the effectiveness of the proposed architecture, an ablation study is applied. Table 3 shows the results obtained with the basic CNN architecture, the results of the mini-ResNET model formed when the convolution layers of the basic CNN architecture are replaced with residual layers, and the results obtained when attention layers are added.

As a result of the ablation study, it was seen that the proposed attention layered mini-ResNET model was more successful on all datasets. Replacing the layers of the Basic CNN architecture with residual layers increased the classification performance significantly. Especially on Dataset 2, which has an unbalanced data distribution, the difference between the results is large. The performance of the basic CNN architecture is close to the mini-ResNet architecture on balanced datasets. The mini-ResNET architecture

**Table 4  Performance of classical machine learning algorithms on permission information.** The values written in bold and underlined style show the accuracy values obtained on the test set.

| Classifier | Dataset 1 | Dataset 2 | Dataset 3 | Dataset 4 | Mix dataset |
|---|---|---|---|---|---|
| | Acc | Acc | Acc | Acc | Acc |
| Naive Bayes | 0.66 | 0.90 | 0.62 | 0.64 | 0.72 |
| Decision Tree | 0.96 | 0.97 | 0.95 | 0.92 | 0.94 |
| KNN | 0.95 | 0.96 | 0.93 | 0.91 | 0.94 |
| SVM | 0.93 | 0.94 | 0.90 | 0.88 | 0.91 |
| LR | 0.94 | 0.97 | 0.95 | 0.95 | 0.91 |
| LDA | 0.93 | 0.97 | 0.94 | 0.84 | 0.91 |
| Random Forest | 0.96 | 0.97 | 0.96 | 0.97 | 0.95 |
| Extra Tree | 0.96 | 0.97 | 0.95 | 0.95 | 0.95 |
| Gradient boosting | 0.94 | 0.97 | 0.95 | 0.95 | 0.92 |
| XGBoost | 0.94 | 0.97 | 0.95 | 0.92 | 0.92 |
| **Proposed method** | **0.97** | **0.98** | **0.98** | **1.00** | **0.97** |

with the attention layer produced more successful results on both balanced and unbalanced datasets.

## DISCUSSION

In this section, the performance of $10 \times 10$ images and $20 \times 20$ images used in the proposed method, the performance of the proposed method against similar permission-based and image-based studies, and the performance against classical machine learning techniques are discussed. In the study, $20 \times 20$ images in which all permission information was used and $10 \times 10$ images created by selecting effective permission information were classified with the proposed architecture using four different datasets and their performance was compared. The classification findings obtained with images created from 100 effective features on all datasets are more successful than $20 \times 20$ images. In CNN networks, the number of parameters plays an important role in factors such as the detection speed of the network, memory consumption and graphics card resource. The proposed attention layer mini-ResNet model contains a total of 399,618 trainable parameters. Although the residual layers and attention layers added to the basic CNN structure cause the number of parameters of the network to increase, the high classification performance achieved makes this disadvantage negligible. Table S3 gives the parameter numbers and input image sizes of well-known transfer learning architectures and the proposed architecture.

Permission information obtained from the datasets used in the study was classified with classical machine learning techniques, and their accuracy performances are presented in Table 4 in comparison with our proposed permission-based method.

The classification performance of the proposed architecture shows that it is more successful than classical machine learning classifiers. The XGBoost algorithm achieved successful results on the datasets with an accuracy rate of 94%, 97%, 95%, 92% and 92%, respectively. However, the 97%, 98%, 98%, 100% and 97% accuracy rates obtained in the proposed architecture make our method superior to classical machine learning techniques.

**Table 5 Performance of similar studies.**

| Author | Dataset | Number of samples | Feature extraction and selection | Classification method | Accuracy |
|---|---|---|---|---|---|
| *Arslan & Tasyurek (2022)* | Drebin, Genome, VirusTotal | 1,920 | Permission 2D-code image | CNN | %96.2 |
| *Yadav et al. (2022)* | R2-D2 | 5,986 | Dex file byte image | EfficientNet-B4 | %95.7 |
| *Zhu et al. (2023)* | Google Play Store, Virusshare | 3,187 | Permission, Hardware and API calls image | MSerNetDroid (CNN) | %96.48 |
| *Şahin et al. (2023)* | – | 2,000 | Permission-Based Features (Lineer Regression) | MLP | %96.1 |
| *Tasyurek & Arslan (2023)* | Drebin, Genome, Arslan's Dataset | 7,721 | Permission RGB image | YoloV5 | %94.2 |
| *Aurangzeb et al. (2024)* | Drebin, Kronodroid, Androzoo | 24,746 | Dex file byte image (PCA) | XGBoost | %95 |
| Proposed method | Androzoo | 40.000 | Permission QR Image (Chi-Square) | Attention layered mini-ResNet | %96.95 |
|  | Arslan's Dataset | 7,622 |  |  | %98.34 |
|  | Drebin, APKPure | 2,000 |  |  | %98.33 |
|  | CICMalDroid 2020, Google Play Store | 500 |  |  | %100 |
|  | Mix Dataset | 50,122 |  |  | %96.78 |

This superiority is provided by the proposed attention layered mini-ResNET architecture as well as the conversion of permission information into images. The results obtained in Table 5 show that more successful results are obtained on the classification performance when the image conversion of permission information is performed. There are classical machine learning and deep learning-based studies using permission information in the literature. Researchers are also interested in image-based studies in Android malware detection to use the feature extraction and classification power of CNNs. Table 5 presents a performance comparison with similar studies using permission information and similar image-based studies.

In performance evaluation, the dataset used, the number of data and data distribution have a great impact. For this reason, four different datasets were used in the experiments to demonstrate the effectiveness of the proposed classification architecture. The number of applications in the datasets varies between 500–40,000. Only in Dataset 2 the data distribution is unbalanced. Data distribution in other datasets is also balanced. In order to show the durability of the classifier against the disadvantages of the datasets, it was used in the Mix Dataset, which consists of the combination of all datasets.

The accuracy performance obtained in the study is between 96.78% and 100% depending on the datasets. Compared to studies that use classical permission information and perform feature selection on permission information, it is seen that our proposed method produces more successful results (*Abdulla & Altaher, 2015*; *Altaher & Barukap, 2017*; *Mat et al., 2022*; *Şahin, Akleylek & Kiliç, 2022*; *Arslan, 2022*; *Şahin et al., 2023*). Dataset 1, where the most unsuccessful results were obtained 96.95% accuracy was achieved with the proposed method. Dataset 1 contains 40,000 applications, and this number is more than the number of applications used by similar studies.

In recent years, image-based studies have become popular for Android malware detection. *Tasyurek & Arslan (2023)* reached 94.2% accuracy with the YoloV5 architecture by converting the permission information into QR code-like 3-channel images. In another

study where they converted images from permission information, they achieved 96.2% accuracy with the CNN-based architecture they proposed (*Tasyurek & Arslan, 2023*; *Arslan & Tasyurek, 2022*). In a similar study, *Zhu et al. (2023)* created images consisting of a matrix of 0 and 1 using API calls and hardware information as well as permission information. It produced an accuracy value of 96.48% with the CNN-based architecture called MSerNet. The number of data and datasets used in these studies, which create images using permission information, are less than our study. At the same time, the results we obtained with our proposed method with different data numbers and different datasets are more successful than studies using permission-based images. There are studies that use byte array for image transformation (*Xiao & Yang, 2020*; *Yadav et al., 2022*; *Aurangzeb et al., 2024*). It has been observed that the proposed method achieves better results compared to studies using byte array images created using QR code-like 10x10 images using permission information.

There are similar studies in the literature that use attention layers together with residual layers (*Zhu et al., 2023*; *Tang et al., 2024*). The architecture created by *Tang et al. (2024)* is an architecture with high-parameter attention layers that works with $256 \times 256$ images. The architecture proposed by *Zhu et al. (2023)* applies the attention mechanism within residual layers. The information from the attention layer is combined with the information from the convolution blocks in the residual layer and is subjected to a convolution process again. The architecture proposed in this study is a minimized 3-block version of the ResNET network. The number of parameters is low and the network can work with images up to $10 \times 10$. At the same time, in the proposed architecture, the information from each residual block enters the attention layer and its output is connected to a residual layer again. This process ensures that features are extracted in each block, that residual features are not ignored, and that specific regions of the feature map are focused on for the next block.

The QR code-like images used in the proposed method consist of effective permission information. The images do not contain meaningless information and are single-channel. On the other hand, the architecture used to classify these images includes both residual and attention blocks according to basic cnn models. Each residual block is followed by an attention block, and in the information sent to the next layer, residual information is not ignored and at the same time, attention is drawn to certain parts of the information map. These factors allow the proposed method to perform better than other methods.

This study has some limitations against the successful performance it exhibits. Permission information may differ in future Android versions. Although the presented attention-layered architecture contains low parameters, it requires a suitable infrastructure (GPU) to operate. Working with deep learning architectures is more complex and computationally more costly than applying classical machine learning techniques.

## CONCLUSION

In this study, the mini-ResNet model with attention layer is proposed for the detection of Android malware. The proposed architecture uses QR code-like $10 \times 10$ images of effective permission information as input images. To ensure the validity of the proposed method, four different datasets were used in the experiments. The datasets contain different

numbers of applications obtained from different sources. Using the chi-square technique on the permission information of the applications, 100 effective permission information were selected and $10 \times 10$ sized QR code-like images were created. A CNN model consisting of five blocks including residual and attention layers was developed for feature extraction and classification. The findings observed as a result of the study are as follows:

- $10 \times 10$ images created from 100 selected features are more successful in classification performance than $20 \times 20$ images.
- The proposed classification model exhibits high accuracy performance on datasets containing a low number of applications and on datasets containing a high number of applications. At the same time, the accuracy values are similar for the balanced and unbalanced dataset.
- It has been observed that the proposed model is successful on the Mix dataset, which was created by combining all datasets, and is resistant to the disadvantages of different datasets.
- When the 10-fold cross validation technique is applied, the results obtained by the proposed model are consistent and the standard deviation values are low.
- As a result of the ablation study, it was observed that attention layers significantly increased accuracy performance.
- The performance of the proposed method in Android malware detection is better than permission information-based studies using classical machine learning.
- The number of parameters of the network is lower than well-known transfer learning architectures and the memory space is smaller.
- It has been observed that the performance of the proposed method is superior to similar studies using permission information images.
- According to the results obtained, the accuracy performance of the attention layered mini-ResNet model in Android malware detection is between 97%–100%.
- Compared to byte array images, images of feature-selected permission information are more effective in classifying malware.

As a result, our proposed method showed successful performance in detecting Android malware. Our method, which achieves better results compared to similar studies, has the ability to make real-time detection with the mobile application or web-based applications to be developed. In future studies, larger feature vectors can be created by using intentions, activities and API call features in addition to permission information, and successful results can be achieved with transformers-based classifiers.

### Funding

The authors received no funding for this work.

### Competing Interests

The authors declare there are no competing interests.

### Author Contributions

- Kazım Kılıç conceived and designed the experiments, performed the experiments, performed the computation work, prepared figures and/or tables, and approved the final draft.
- İbrahim Alper Doğru analyzed the data, prepared figures and/or tables, authored or reviewed drafts of the article, and approved the final draft.
- Sinan Toklu conceived and designed the experiments, analyzed the data, authored or reviewed drafts of the article, and approved the final draft.

### Data Availability

Dataset 1 is available at Kaggle: https://www.kaggle.com/datasets/kazimkili/dataset1.

Dataset 2 is available at: DOI: 10.7717/peerj-cs.533/supp-1; DOI: 10.7717/peerj-cs.533/supp-2.

The code and data are available in the Supplemental Files.

### Supplemental Information

Supplemental information for this article can be found online at http://dx.doi.org/10.7717/peerj-cs.2362#supplemental-information.

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
