# Peer review of "PermQRDroid: Android malware detection with novel attention layered mini-ResNet architecture over effective permission information image"

_PeerJ Computer Science, doi:10.7717/peerj-cs.2362_

## Round 0.1 · original submission · Major Revisions

The review process is now complete. While finding your paper interesting and worthy of publication, the referees and I feel that more work could be done before the paper is published. My decision is therefore to provisionally accept your paper subject to major revisions.

Reviewer 1 ·

Basic reporting

The researchers proposed an attention-layered mini-ResNet model for detecting Android malware. They created QR code-like images using effective permission information from various datasets and achieved high classification accuracy with their novel architecture, which combines residual and attention layers to enhance performance.

Language and Clarity: The manuscript is generally well-written but could benefit from minor language refinements for clarity. Consider revising sentences that are overly complex or verbose. For example, the sentence "The success and durability of the proposed method in different environments have been tested through experiments conducted on different datasets" could be simplified to "The proposed method's success and durability were tested using various datasets."

Literature Review: The literature review is thorough and well-cited. However, some sections could benefit from a more critical analysis of previous works. This would help in highlighting the novelty and importance of your study.

Figures and Tables: Figures and tables are relevant and generally of high quality. Ensure that all figures are labeled clearly and described in the text. For instance, Figure 1 could benefit from a more detailed caption explaining its components and relevance.

Experimental design

Methodology Clarity: While the methods are described in detail, the paper would benefit from a more structured methodology section. Consider breaking down the methodology into sub-sections with clear headings such as "Data Collection," "Feature Extraction," "Model Architecture," and "Training Procedure." This will help readers follow your experimental design more easily.

Data Description: Provide more detailed descriptions of the datasets used, including their sources, sizes, and any preprocessing steps taken. This information is crucial for replicability.

Parameter Justification: Justify the choice of specific parameters used in your model, such as the number of layers in the Mini-ResNet, the size of the attention layer, and the learning rate. Explain why these parameters were chosen and how they impact the model's performance.

Validity of the findings

Statistical Analysis: Ensure that all statistical analyses are robust and clearly explained. For example, when discussing the accuracy of the model, include confidence intervals or standard deviations to provide a sense of variability and reliability of your results.

Comparative Analysis: While the results are compared with previous studies, this section could benefit from a more in-depth discussion. Highlight the strengths and weaknesses of your approach compared to existing methods, and explain why your method performs better.

Limitations: Discuss the limitations of your study in more detail. For example, mention any potential biases in the datasets used, or any assumptions made in the model that could affect its generalizability.

Reviewer 2 ·

Basic reporting

The paper proposes a mini-ResNet model that can detect malware in Android applications using QR code-like images created from permission information.

- The figures and tables are of quality and well-labeled with relevant information. However, some figures could benefit from more descriptive titles and captions.
- Many sentences are unclear and confusing. For example, the sentence "The popularity of the Android operating system, which is popular with users, is increasing day by day..." contains unnecessary repetition and lacks clarity. The language should be checked.
- The introduction does not adequately provide the context of the study. The popularity and security issues of the Android operating system have been addressed superficially without in-depth analysis.
- The study's place in the literature and its original contributions are not sufficiently emphasized. Contributions should be detailed in a separate subsection.
- There has been a significant number of studies in this field in recent years. Therefore, the review of the relevant literature appears to be incomplete.

Experimental design

- Some steps need more detailed explanation. In particular, the process of creating QR code-like images should be elaborated on, as it is not clear which algorithms were used in this process.
- The research question is not clearly defined, and the aim of the study is ambiguous. It is not explained how this study fills the knowledge gap in the literature, which reduces the importance of the study.
- How the chi-square test was applied and how the selected features were determined should be explained in more detail. The current explanations are insufficient and unclear.

Validity of the findings

- The results and discussion sections are quite superficial. More detail is needed.
- The number of data points for Dataset 4 is small, and the success rate is 100%. This does not provide sufficient information about this part of the study.
- Datasets like Androzoo, Drebin, and CICMalDroid have been used many times. Therefore, the differences from other studies should be highlighted in more detail.
- The contribution or impact of converting the dataset into images should be provided in a separate section in more detail.

Reviewer 3 ·

Basic reporting

In this paper, the authors propose an image-feature based method for Android malware detection and design an attention-based CNN network for the classification stage. However, there are major concerns as follows:

- In the contribution section of the study, the authors present the following points as innovative aspects. However, it is not accurate to consider these as contributions to the literature.

"In this study, 4 datasets were obtained from different sources to verify the proposed method. More than 40,000 applications were collected for three of these datasets and features were extracted using the static analysis method."
"All datasets were combined by selecting 100 effective permission information, and the durability of the proposed model against the disadvantages and imbalance problems of these different datasets was tested."

- The authors claim that an innovative CNN network is introduced in the paper using an attention module and ResNet model. However, when reviewing the literature (the relevant study is presented below), it is evident that a similar CNN model structure has already been proposed. The innovative aspect of the current study in terms of model structure (less conv. layer?) is therefore not entirely clear.

[1] Tang, J., Xu, W., Peng, T., Zhou, S., Pi, Q., He, R., & Hu, X. (2024). Android malware detection based on a novel mixed bytecode image combined with attention mechanism. Journal of Information Security and Applications, 82, 103721.

- The reference provided for the attention module (Niu et al., 2021) may not be accurate. It would be more appropriate to reference the original paper.

Experimental design

- To ensure the consistency of the results obtained in the classification stage, it would be better to partition the dataset using 10-fold cross-validation and provide the mean accuracy and standard deviation values. The results obtained with the 70-15-15 partitioning given in the study should be considered in relation to how the dataset was partitioned. This is particularly evident from the 100% accuracy results obtained for some test data.

- To better understand the contribution of the attention module, it would be beneficial to include an ablation study. For this purpose, it is important to remove the attention module from the CNN model, train it on the same dataset, and share the test results. This will help to understand better the contribution of the proposed model to the literature.

Validity of the findings

- When examining the training-validation curve results presented in Figure 6 (a), (b), and (e), it is observed that the model starts to overfit. This can be mitigated by considering the learning rate or batch size structure used during the training phase. Additionally, the use of cross-validation can also be considered as a solution to this problem.

- The CNN model used and the training phase should be presented in more detail, including the number of convolution kernels, stride, kernel size, the total number of epochs, etc.

- Adding the results obtained from the AUC evaluation metric is essential for a better assessment of the results.

- The proposed method is compared with some studies in the literature. In addition to providing the number of samples, it is important to also mention the names of the datasets. This will allow for a better understanding of how the proposed method differs from similar studies using similar datasets.

---

## Round 0.2 · accepted · Accept

Since the reviewers' comments have been addressed and the reviewers recommend acceptance, we are happy to inform you that your paper has been accepted for publication.

Reviewer 1 ·

Basic reporting

Authors updated the paper and no further update needed from my side.

Experimental design

As above

Validity of the findings

As above

Additional comments

As above

Reviewer 2 ·

Basic reporting

The authors have made the necessary revisions.

Experimental design

-

Validity of the findings

-

Reviewer 3 ·

Basic reporting

The authors have successfully completed the requested revision.

Experimental design

The authors have successfully completed the requested revision.

Validity of the findings

The authors have successfully completed the requested revision.